# Characteristics of Vegetation Change and Its Climatic and Anthropogenic Driven Pattern in the Qilian Mountains

Yanmin Teng [1] 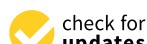, Chao Wang [2], Xiaoqing Wei [3], Meirong Su [4,*], Jinyan Zhan [5,*] and Lixiang Wen [5]

1   Research Center for Eco-Environmental Engineering, Dongguan University of Technology, Songshan Lake, Dongguan 523808, China; tengym@dgut.edu.cn
2   School of Labor Economics, Capital University of Economics and Business, Beijing 100089, China; wangc@cueb.edu.cn
3   School of Energy and Water Resources, Shenyang Institute of Technology, Shenyang 113122, China
4   Key Laboratory for City Cluster Environmental Safety and Green Development of the Ministry of Education, School of Ecology, Environment and Resources, Guangdong University of Technology, Guangzhou 510006, China
5   State Key Laboratory of Water Environment Simulation, School of Environment, Beijing Normal University, Beijing 100875, China
*   Correspondence: sumr@gdut.edu.cn (M.S.); zhanjy@bnu.edu.cn (J.Z.)

**Abstract:** The Qilian Mountains (QLM) are an essential ecological security barrier in northwest China. Identifying the driven pattern of vegetation change is crucial for ecological protection and restoration in the QLM. Based on high-resolution vegetation coverage (VC) data in the QLM from 1990 to 2018, linear trend analysis was employed to examine the spatiotemporal dynamics of VC in the QLM, while correlation analysis was utilized to establish relationships between VC change and environmental factors. Multiple correlation analysis and residual analysis were adopted to recognize the climatically and anthropogenically driven pattern of VC change. The results showed that VC in the QLM presented a remarkable upward trend in volatility from 1990 to 2018. The significant increase areas accounted for 59.32% of the total, mainly distributed in the central and western QLM, and the significant decrease areas accounted for 9.18%, mostly located in the middle and eastern QLM. VC change showed a significant positive correlation with precipitation change and annual average temperature, while it exhibited a significant negative correlation with annual average precipitation, current VC status, livestock density, and slope. Climate change played a leading role in the increase of VC, and the impact of precipitation was significantly higher than that of temperature. Affected by climate change, the VC of alpine steppes and temperate steppes increased the most. Under the human interference, VC decreased significantly in 9.2% of the region, of which shrubs fell the most, followed by alpine meadows and forests. This study can provide certain guidance for local ecological protection and restoration efforts.

**Keywords:** vegetation coverage; climate change; human activities; vegetation degradation; Qinghai–Tibetan Plateau



## 1. Introduction

As a natural link between soil, atmosphere, and water, vegetation plays an important role in terrestrial ecosystems by regulating carbon cycling and energy exchange [1,2]. Vegetation coverage (VC) can reflect the regional comprehensive characteristics of climate, topography and human activities, and is easily affected by climatic and anthropogenic factors [3,4]. Therefore, it is an important indicator for monitoring the impact of climate change and human activities on ecosystems [5,6], and it has been widely used in large-scale ecological environment change monitoring and assessment [7–9].

In terms of identifying VC change, researchers mostly adopt linear regression trend analysis [10–14], and a few adopt the ordinary least squares method [15]. At present the,

GeoDetector model is one of the most commonly used methods to detect the drivers of vegetation change and has received extensive attention from researchers [16–18]. GeoDetector has many advantages, including the ability to analyze both numerical and qualitative data, as well as to identify the interaction between two driving factors [10,19,20]. However, it can only analyze the influence intensity of driving factors statistically and is unable to display the spatial heterogeneity of driving factors' influence on a map [12,21]. Hence, studies have usually adopted correlation analysis (or partial correlation analysis) to spatially clarify the driving effect of climate variables on vegetation change [11,13]. In addition, the geographic weighted regression model is also frequently used to spatially identify the comprehensive driving effects of multiple factors [15,22]. As for human factors, residual analysis is often adopted to spatially identify the influences of human intervention on vegetation change [14,23].

Among the climatic factors, temperature and precipitation are the most important [15,24,25]. The effects of climatic factors on VC change differ spatially, and precipitation dominates in arid and semi-arid regions, while temperature dominates in humid southern regions [26]. For the Qinghai–Tibet Plateau (QTP), precipitation plays a decisive role in VC change in the Three River Source Region, Heihe River Basin, and Yarlung Zangbo River Basin [10,21,27]. However, studies also show that annual mean temperature is the dominant factor driving VC change in the northeastern Tibetan Plateau [12,28]. In addition to precipitation and temperature, solar radiation and wind speed are also important driving factors affecting vegetation change [29]. Moreover, elevation and soil type were found to be the main factors affecting VC in the Qilian Mountains (QLM) [17,30]. For elevation gradient, VC in the QLM increased first and then decreased with the increase of altitude [30]. From the long-term change, VC showed an increasing trend at low altitudes (below 3200 m) and gentle slopes (below 15°), and a decreasing trend at high altitudes (above 3700 m) and steep slopes (above 25°) [31]. For soil type, black felt soil has abundant humus, which can provide rich nutrients for vegetation [17]. Human activities also have a significant impact on VC change, especially in the semi-humid region of China [19]. Land use change is a dominant factor, especially in semi-humid, semi-arid, and arid areas [21,32]. In addition, GDP and population density also have a certain impact [10].

Within the QTP, climate change in the QLM is more significant, and the vegetation is more susceptible to climate change as well [12,33]. Since the 1980s, VC in the QLM has shown an overall improvement trend, with significant increases in the central and western regions and partial degradations in the central and eastern regions [34–37]. However, due to different analysis periods, there are significant differences in the degraded areas identified by several studies. In addition, studies further analyzed the correlation between vegetation change and climate change in the QLM and found that precipitation was the main factor leading to vegetation change, followed by temperature [34–37]. Moreover, the correlation between VC and temperature in winter and spring was the highest, and the correlation between VC and precipitation in summer was the highest in the previous period [35].

Climate change has promoted VC increase in the QLM on a large scale. However, due to the impacts of human activities such as overgrazing, mining, and hydropower development, vegetation degradation and land desertification in local areas of the QLM were prominent in recent decades [38,39], thus affecting the overall role of the ecological security barrier [40]. To control the degraded lands and restore regional ecological functions, the local government has implemented a series of ecological protection and restoration projects in the QLM, such as a grazing-forbidding project, a grassland–livestock balance project, and a grassland ecological award and compensation project [41,42]. These projects reduced livestock numbers to a certain extent, increased the construction of ecological engineering projects including fences and artificial grasslands, and promoted vegetation restoration [43]. However, the climatically and anthropogenically driven patterns of VC change in this region are not yet clear. Therefore, we aim to clarify the temporal and spatial characteristics of VC change in the QLM and recognize its main natural and anthropogenic

influencing factors. Further, we will identify the VC change zones driven by climate change and human activities, as well as the VC change types under different vegetation types. This study has a certain guiding significance for targeted implementation of ecological protection and restoration measures in the QLM.

## 2. Materials and Methods

### 2.1. Study Area

The QLM (93°30′–103°00′ E, 35°43′–39°36′ N) is located at the northeast edge of the Qinghai–Tibet Plateau, with a total area of about 193,300 km$^2$. It is the source of Heihe River, Datong River, Huangshui River, Shule River, and Shiyang River (Figure 1a). The QLM is high in the central and northwest and low in the southeast, and most of the areas are about 3500–5000 m above sea level (Figure 1a). The QLM has a typical continental climate: the eastern part is warm and humid, having a continental semi-arid alpine grassland climate, and the western part is cold and dry, with a continental arid desert climate [44]. The main vegetation types included alpine meadow, alpine steppe, temperate steppe, temperate desert, alpine desert, and shrub, in addition to small areas of coniferous forest, broad-leaved forest, and cropland (Figure 1b). Based on previous studies [45,46], we obtained the characteristics of different vegetation types and their dominant plant species: the main tree species for forests are Simon poplar (*Populus simonii* Carr.), white birch (*Betula platyphylla* Suk), aspen (*Populus davidiana* Dode), Qinghai spruce (*Picea crassifolia*), Chinese pine (*Pinus tabuliformis*), and Qilian juniper (*Juniperus przewalskii* Kom.); shrubs are led by cold-tolerant shrubs, including *salix gilashanica*, *caragana jubata*, bush cinqefoil (*Potentilla fruticosa* L.), and sea buckthorn (*Hippophae rhamnoides* L.); as a good summer pasture, alpine meadow develops under moderate moisture conditions and is dominated by perennial mesophytes (such as *Kobresia*); alpine steppe is mainly composed of cold- and drought-resistant perennial herbs (such as *Carex*) and cushion plants; temperate steppe is dominated by herbs suitable for warm and arid climates (such as *Stipa*); alpine desert is largely composed of cold and xerophytic shrubs, accompanied by a certain number of cushion herbaceous plants; and temperate desert is dominated by xerophytic tufted grasses and shrubs. The hydrothermal regime in the QLM varies greatly, with an annual mean precipitation of 67–758 mm and annual mean temperature of −16.3–6 °C (Figure 1c,d). Affected by climatic conditions, VC in the QLM shows a significant downward trend from east to west (Figure 1e). Similarly, the population is concentrated in the lower elevations of the eastern QLM (Figure 1f).

### 2.2. Data Sources

Monthly and annual meteorological data from 1990 to 2018 were obtained from the National Tibetan Plateau Data Center (TPDC, https://data.tpdc.ac.cn (accessed on 8 May 2022)), including temperature and precipitation, with a spatial resolution of 1 km. These data were generated from the global high-resolution climate dataset published by WorldClim through the Delta spatial downscaling scheme [47]. VC data of seven periods (1990, 1995, 2000, 2005, 2010, 2015, and 2018) at 30 m spatial resolution were provided by the TPDC (accessed on 8 March 2020). Annual VC data at 1 km spatial resolution from 1990 to 2018 were derived from the Resource and Environmental Science and Data Center (RESDC, https://www.resdc.cn (accessed on 19 April 2022)). Vegetation type data with a 1 km spatial resolution and rural settlement shape files were obtained from the RESDC (accessed on 24 October 2020). A digital elevation model (DEM) product with a spatial resolution of 30 m was provided by Qi et al. [48] (accessed on 4 July 2021). Road networks data were obtained from the TPDC (accessed on 15 September 2021) and derived from a 1:100,000 ADC_WorldMap (2014), including major highways, roads, and railways. Grazing intensity data in 2010 with a 1 km spatial resolution were acquired from the Food and Agriculture Organization of the United Nations (http://www.fao.org/geonetwork/srv/en/main.home (accessed on 19 May 2020)). Population density data in 2015 at 1 km

resolution were obtained from the Chinese population spatial distribution dataset provided by the RESDC (accessed on 4 July 2021).

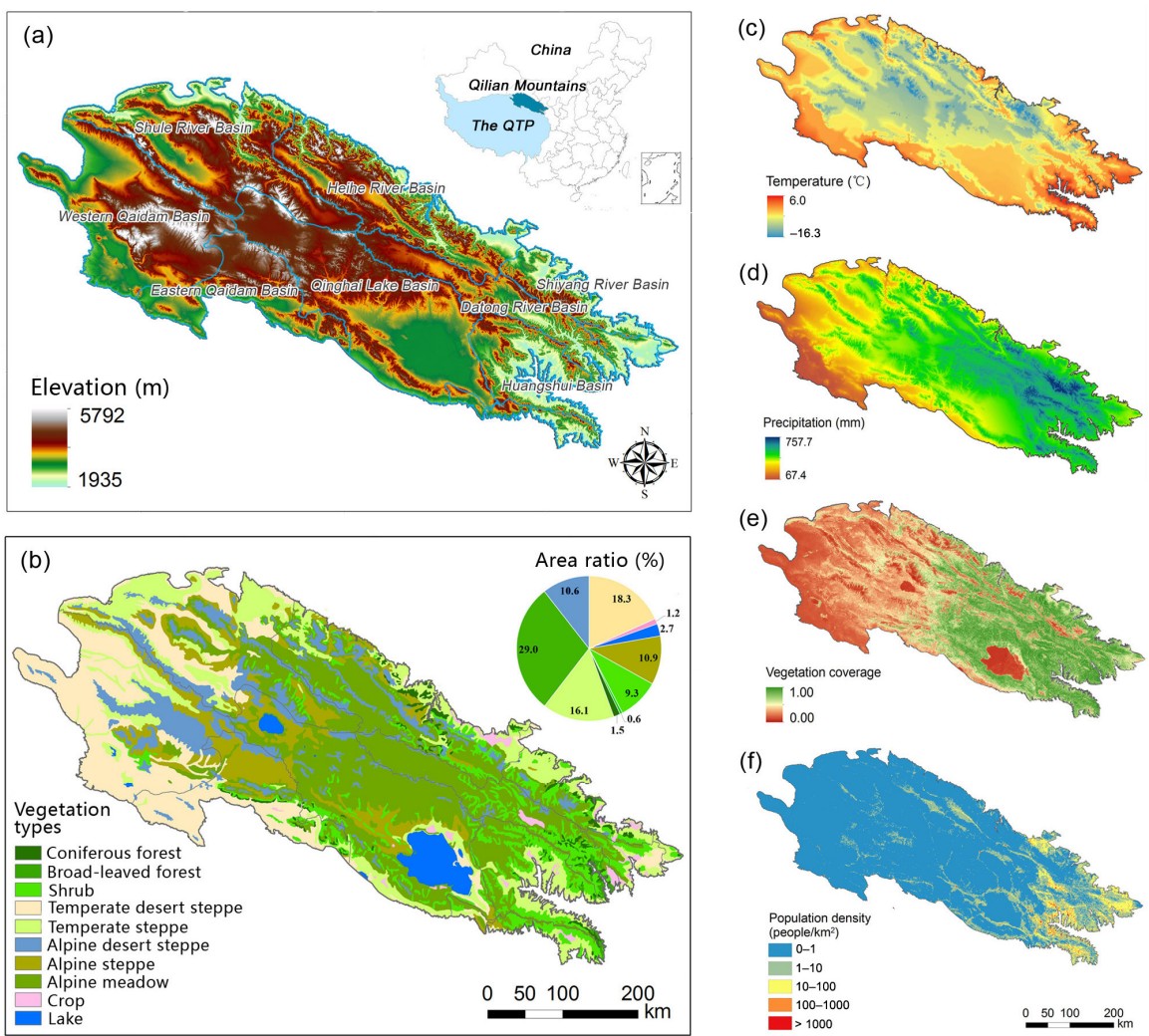

**Figure 1.** Basic information of the study area. (**a**) Location of the QLM and a DEM map with basin boundaries, (**b**) vegetation types, (**c**) temperature, (**d**) precipitation, (**e**) vegetation coverage, and (**f**) population density.

*2.3. Research Methods*

2.3.1. Change Trends of VC, Temperature and Precipitation

Based on the seven-period, 30 m resolution VC data, excluding cultivated land and non-vegetated areas such as desert and water bodies, we identified the high-resolution spatial pattern of vegetation change in the QLM. The change trends of VC, temperature, and precipitation at a grid scale over the QLM were analyzed by using linear regression trend analysis. After the data were preprocessed, trend analysis and significance testing of each factor were conducted via MATLAB (R2019b). The main calculation formula is as follows [49]:

$$S = \frac{n \times \sum_{i=1}^{n} i \times X_i - (\sum_{i=1}^{n} i) \times (\sum_{i=1}^{n} X_i)}{n \times \sum_{i=1}^{n} i^2 - (\sum_{i=1}^{n} i)^2} \tag{1}$$

where $S$ is the change rate of VC, temperature, and precipitation; $n$ is the number of years. A positive value indicates an overall upward trend, while a negative value indicates an overall downward trend.

Further, based on the boundary data of the QLM, we extracted the average values of 1 km resolution annual VC, temperature, and precipitation year by year in ArcGIS. Then, we used the extracted data to create line plots and obtained the change trends of VC, temperature, and precipitation in time series from 1990 to 2018.

### 2.3.2. Correlation Analysis between VC Change and Climate Change

Pearson correlation analysis was used to determine the correlations between VC change and temperature and precipitation change. The main calculation methods are as follows:

$$R = \frac{\sum_{i=1}^{n}(X_i - \overline{X})(Y_i - \overline{Y})}{\sqrt{\sum_{i=1}^{n}(X_i - \overline{X})^2 * \sum_{i=1}^{n}(Y_i - \overline{Y})^2}} \tag{2}$$

where $R$ is the correlation coefficient; $n$ is total research years; $X_i$ is VC of the year $i$; $Y_i$ is temperature and precipitation of the year $i$; and $\overline{X}$ is the mean VC from 1990 to 2018. $\overline{Y}$ is the average temperature or precipitation from 1990 to 2018. When $R > 0$, each factor is positively correlated, when $R < 0$, each factor is negatively correlated, and the correlation increases as the $R$ value approaches 1 or $-1$.

Partial correlation analysis can reveal the impact intensity of one factor on another while other factors remain unchanged [11,13]. Multiple correlation analysis can analyze the correlation between two or more factors and a certain factor, so as to identify the comprehensive influences of multiple factors on this factor [50]. In this study, partial correlation and multiple correlation analysis of VC, temperature, and precipitation were carried out to identify the dominant areas of VC change affected by different climatic factors in the QLM. The calculation formulas of the partial correlation analysis and $t$-test are as follows:

$$R_{xy,z} = \frac{R_{xy} - R_{xz}R_{yz}}{\sqrt{(1 - R_{xz}^2) * (1 - R_{yz}^2)}} \tag{3}$$

$$t = \frac{R_{xy,z} * \sqrt{n - m - 1}}{\sqrt{1 - R_{xy,z}^2}} \tag{4}$$

where $R_{xy,z}$ is the partial correlation coefficient between the dependent variable $x$ and the independent variable $y$ when the independent variable $z$ remains constant; $R_{xy}$, $R_{xz}$, and $R_{yz}$ are correlation coefficients between factors $x$, $y$, and $z$; $n$ is sample size; and $m$ is the degree of freedom.

The calculation formulas of the multiple correlation analysis and F-test are as follows:

$$R_{x,yz} = \sqrt{1 - (1 - R_{xy}^2) * (1 - R_{xz,y}^2)} \tag{5}$$

$$F = \frac{R_{x,yz}^2}{1 - R_{x,yz}^2} \frac{n - m - 1}{m} \tag{6}$$

where $R_{x,yz}$ is the multiple correlation coefficient between the dependent variable $x$ and the independent variables $y$ and $z$; $n$ is sample size; and $m$ is the degree of freedom.

### 2.3.3. Identification of Climate-Driven Zones

With reference to previous research [51], we formulated the zoning criteria for driving factors of VC change in the QLM (Table 1). The climate-driven types include the temperature-driven zone, precipitation-driven zone, temperature and precipitation co-driven zone, and non-climate-driven zone.

**Table 1.** Zoning criteria for driving factors of VC change.

| VC Change Types | Zoning Criteria | | |
|---|---|---|---|
| | $R_{xy,z}$ | $R_{xz,y}$ | $R_{x,yz}$ |
| Temperature-driven type | $t \geq t_{0.05}$ | | $F \geq F_{0.05}$ |
| Precipitation-driven type | | $t \geq t_{0.05}$ | $F \geq F_{0.05}$ |
| Co-driven type | $t \leq t_{0.05}$ | $t \leq t_{0.05}$ | $F \geq F_{0.05}$ |
| Non-climate driven type | | | $F \leq F_{0.05}$ |

Note: $R_{xy,z}$ and $R_{xz,y}$ represent partial correlation coefficients between VC and temperature and precipitation, respectively; $R_{x,yz}$ represents the multiple correlation coefficient between VC and climatic factors; $t$ and $F$ represent the statistical values of the $t$-test and F-test, respectively; 0.05 is the significance level.

### 2.3.4. Residual Analysis of VC Change

Residual analysis is a quantitative analysis method to analyze the impact of human activities on VC change [52] and is widely used in large-scale vegetation change analysis [53,54]. Based on temperature and precipitation data, multivariate linear regression was performed to fit the predicted VC, that is, the expected VC only under the influence of climatic factors. The effect of anthropogenic factors on vegetation can be obtained by subtracting the real VC value from the predicted value. The calculation formula is as follows:

$$VC_{\text{predicted}} = a \times T + b \times P + c \tag{7}$$

$$\varepsilon = VC_{\text{real}} - VC_{\text{predicted}} \tag{8}$$

where $VC_{\text{predicted}}$ and $VC_{\text{real}}$ are the predicted VC value based on regression models and the observed VC value based on VC data with a resolution of 30 m, respectively; $T$ and $P$ are temperature and precipitation, respectively; $a$, $b$, and $c$ are model parameters; and $\varepsilon$ is the residual, that is, the impacts of human activities on VC. When $\varepsilon > 0$, human activities promote the increase of VC, and when $\varepsilon < 0$, human activities cause the decrease of VC. The greater the absolute value, the stronger the influences of human activities on vegetation.

## 3. Results

### 3.1. Change Features of VC

From 1990 to 2018, VC in most areas of the QLM showed an increasing trend, with 47.05% and 12.27% of the areas significantly and extremely significantly increasing in VC, respectively (Figure 2a). At the same time, VC decreased in local areas, with 7.54% and 1.64% of the areas significantly and extremely significantly decreasing in VC, respectively. For the spatial distribution, areas with significant increase in VC were mainly distributed in the western QLM with weak human disturbances, mainly involving the Shule River Basin, Qinghai Lake Basin and the eastern and western sections of Qaidam Basin. Areas where VC decreased remarkably were mostly located in the central and eastern QLM with high human activity intensity, mainly involving the Heihe River Basin, Datong River Basin, Huangshui River Basin, Shiyang River Basin, and Qinghai Lake Basin. In terms of time dynamics, VC in the QLM showed a significant fluctuating upward trend from 1990 to 2018, with an average annual growth rate of about 0.55% (Figure 2b).

Furthermore, we extracted the mean VC change rate under different vegetation types (Figure 2c). The results showed that the change rates of alpine and temperate grasslands were positive, with values of 0.97% and 0.47%, respectively. On the contrary, the VC of forests, shrubs, and alpine meadows showed a decreasing trend, with shrubs showing the largest change (−0.18%), followed by alpine meadows (−0.15%), and forests (−0.06%). Hence, VC increase in the QLM was mainly dominated by alpine grassland and temperate grassland, and forests, shrubs and alpine meadows degraded to some extent under human interference.

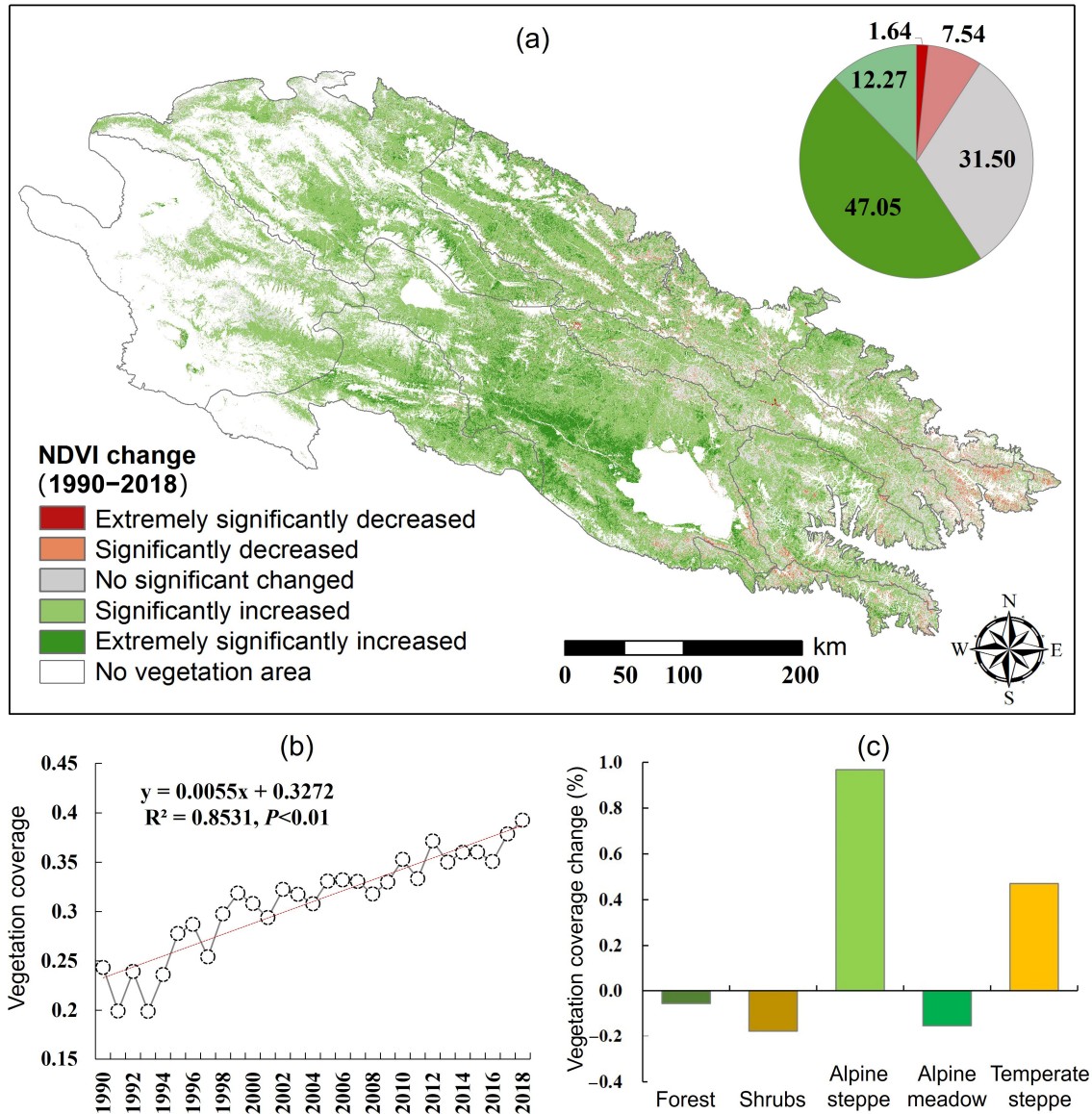

**Figure 2.** Vegetation coverage (VC) change in the QLM from 1990 to 2018. (**a**) Spatial variation, (**b**) temporal change, and (**c**) changes under different vegetation types.

We extracted the VC change rate and various environmental variables and then conducted Pearson correlation analysis to recognize the main influencing factors of VC change. The results showed that VC change was significantly negatively correlated with elevation, slope, livestock density, current VC status, annual mean precipitation, and precipitation change rate, while it was significantly positively correlated with annual mean temperature, distance from roads, and distance from rural settlements (Table 2). Among them, the correlation between VC change rate and annual mean precipitation was the highest (−0.583), followed by VC, livestock density, precipitation change rate, annual mean temperature, slope, distance from roads, distance from rural settlements, elevation, and temperature change rate. Therefore, areas with significant vegetation degradation tend to be distributed in areas with good ecological environments (high precipitation and VC, and obvious precipitation increase) and high human activity intensity (high livestock density and close to rural settlements and roads).

**Table 2.** Correlation between VC change and environmental factors.

| Environmental Factors | Elevation | Slope | Distance from Rural Settlements | Distance from Roads | Livestock Density |
|---|---|---|---|---|---|
| Correlation coefficient | −0.022 * | −0.167 ** | 0.064 * | 0.079 * | −0.265 ** |
| **Environmental Factors** | **Current Vegetation Coverage Status** | **Annual Mean Temperature** | **Annual Mean Precipitation** | **Temperature Change Rate** | **Precipitation Change Rate** |
| Correlation coefficient | −0.327 ** | 0.175 ** | −0.583 ** | −0.013 | −0.257 ** |

Note: * represents $p < 0.05$, ** represents $p < 0.01$.

### 3.2. The Impact of Climate Change on VC Change

From 1990 to 2018, temperature and precipitation showed a significant increase trend, with average annual increases of about 0.027 °C (Figure 3a) and 2.38 mm (Figure 3b), respectively. Regarding the spatial pattern, the regions with obvious temperature and precipitation increases were mainly located in the northwest QLM (Figure 3a) and the high-altitude area in the central QLM (Figure 3b), respectively. The impact of temperature change on VC change is relatively small and exhibits a scattered distribution feature (Figure 3c). Approximately 8.6% and 5.4% of the areas had significant positive and negative correlations between temperature change and VC change, respectively. In contrast, the impact of precipitation change on VC change is more significant, and it is concentrated in the central QLM (Figure 3d). About 45.2% and 1.2% of the areas showed significant positive and negative correlations between precipitation change and VC change, respectively.

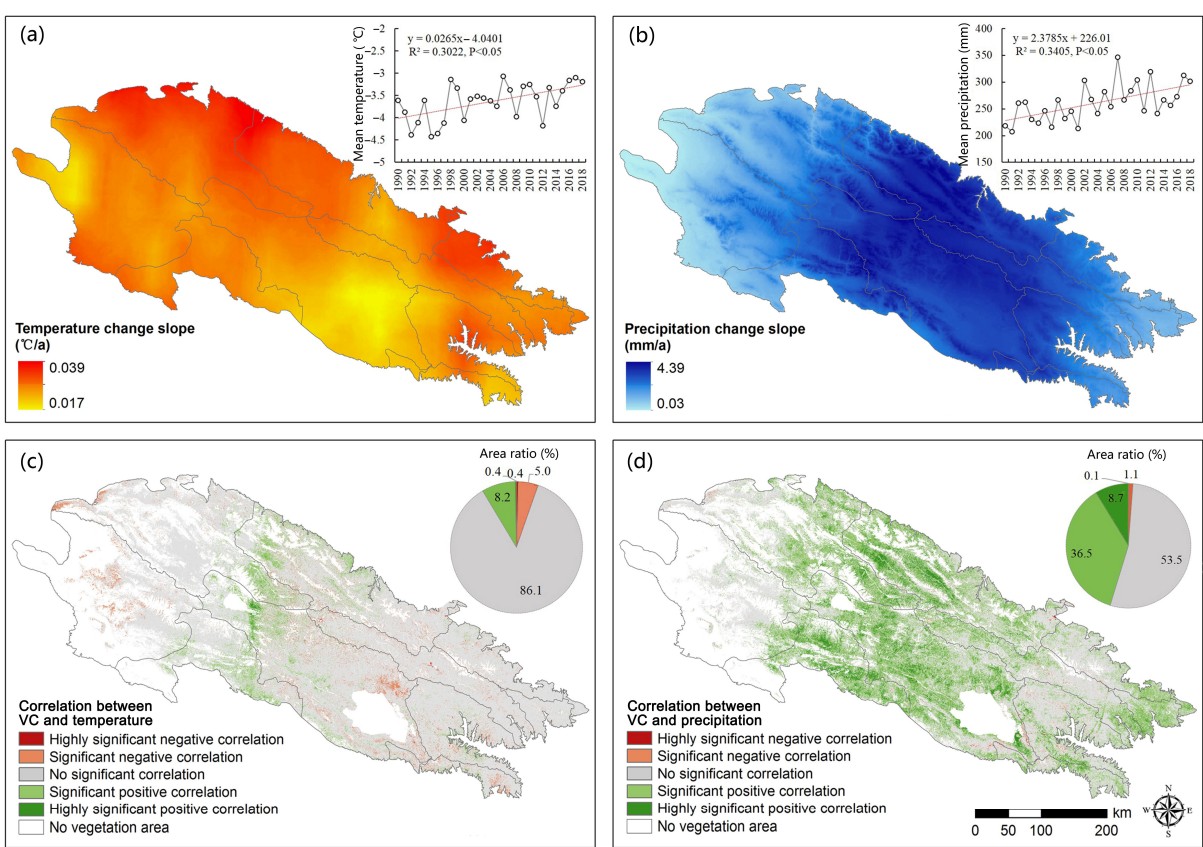

**Figure 3.** Temperature change (**a**), precipitation change, (**b**) and their correlation (**c**,**d**) with vegetation coverage (VC) change in the QLM. The significant and highly significant correlations are the situations under $p < 0.05$ and $p < 0.01$, respectively.

The partial correlation analysis results were consistent with those of the correlation analysis, but the effect of solely temperature or precipitation was slightly stronger (Figure 4a,b). The multiple correlation analysis shows that 54.2% and 5.4% of the regional VC changes in the QLM were significantly and extremely significantly positively correlated with temperature and precipitation changes, respectively (Figure 4c). Therefore, the increases of temperature and precipitation were the main factors leading to the VC increase in the QLM.

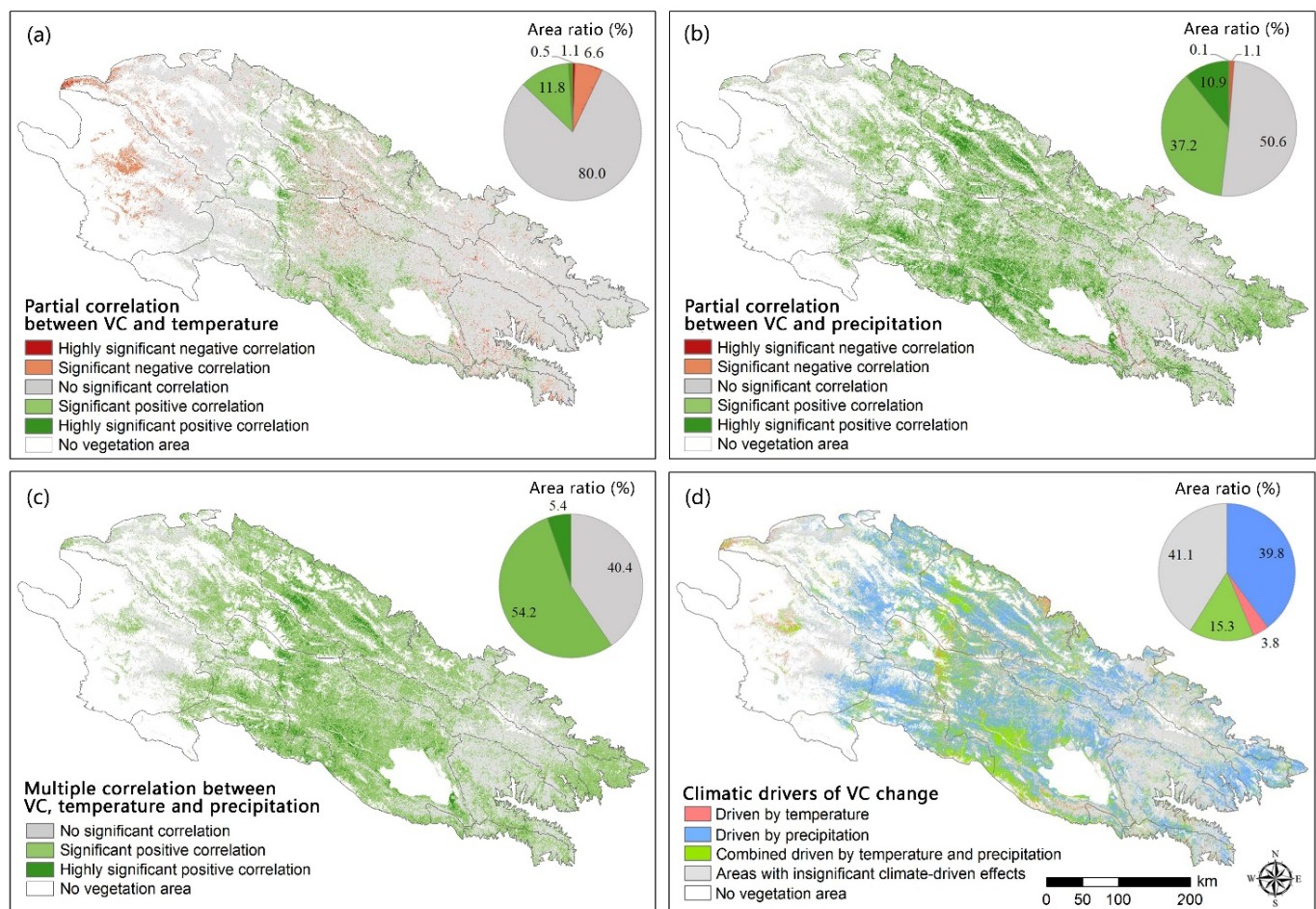

**Figure 4.** Partial correlation (**a**,**b**) and multiple correlation (**c**) between vegetation coverage (VC) change and climate change, as well as climate-driven patterns (**d**) of VC change in the QLM. The significant and highly significant correlations are the situations under $p < 0.05$ and $p < 0.01$, respectively.

Based on the results of partial and multiple correlation analysis, the climate-driven types of VC change in the QLM were identified, including the temperature-driven zone, precipitation-driven zone, temperature and precipitation co-driven zone, and non-climate-driven zone (Figure 4d). The climate-driven zone accounted for 58.9% of the total QLM. Among the zones, the precipitation driven zone accounted for 39.8% of the total, and was largely distributed in the central high-altitude region and the eastern edge of the QLM. The co-driven zone occupied 15.3%, mostly being distributed in the central QLM. The temperature-driven zone only accounted for 3.8%, and the distribution was scattered.

### 3.3. Effects of Human Activities on VC Change

The areas where human activities had an obvious negative effect on VC change accounted for 17.3% of the study area, mainly distributed in the southeastern QLM, especially in the lower reaches of Huangshui Basin, Qinghai Lake Basin, and Datong River Basin (Figure 5a,b). The areas where human activities played a significant positive role accounted

for 17.7%, mainly distributed in the midwestern and northeastern QLM, especially in the western part of the Heihe River Basin and Qaidam Basin, as well as the upper reaches of Qinghai Lake Basin and Shule River Basin. Spatially, the regions where the residuals decreased remarkably were largely distributed in the central and northern QLM, involving the Heihe River Basin, Datong River Basin and Shiyang River Basin (Figure 5c). In terms of dynamic changes, the proportions with significant reductions and increases in residuals were 8.6% and 14.8%, respectively (Figure 5d). Hence, the vegetation degradation area caused by human disturbances was smaller than the ecological restoration area in the QLM.

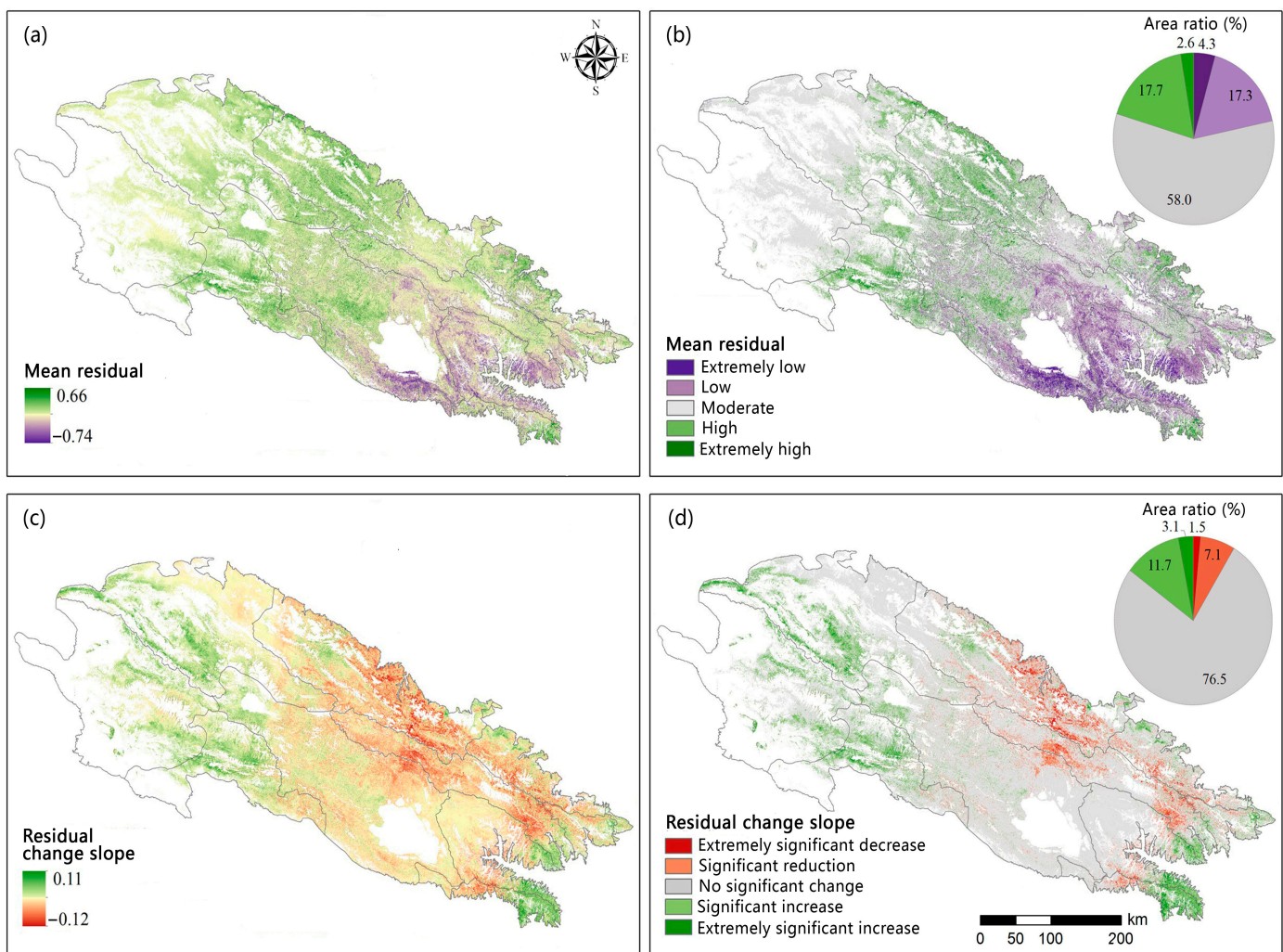

**Figure 5.** Spatial distribution of mean residual (**a**,**b**) and residual slope (**c**,**d**) of vegetation cover change in the QLM.

### 3.4. Climatically and Anthropogenically Driven Pattern of VC Change

The climatically and anthropogenically driven pattern (Figure 6) of VC change in the QLM during 1990–2018 was obtained by superimposing the VC change map (Figure 2), the complex correlation map (Figure 4c), and the residual map (Figure 5a). Climate change played a significant role in the improvement of VC, and the area driven by climate change is 37.9% of the total. In addition, climate change and human activities together driving the increase of the VC area accounted for 10.7% of the total. Within these regions, human activities further promoted vegetation restoration with the contribution of climate change. Therefore, VC increased significantly in nearly half of the QLM under the influence of climate change. However, 9.2% of the areas showed a significant VC decrease under the interference of human activities. In addition, 10.6% of the areas showed a significant

improvement in VC, and were largely distributed in the western part of the QLM where human activities were weak.

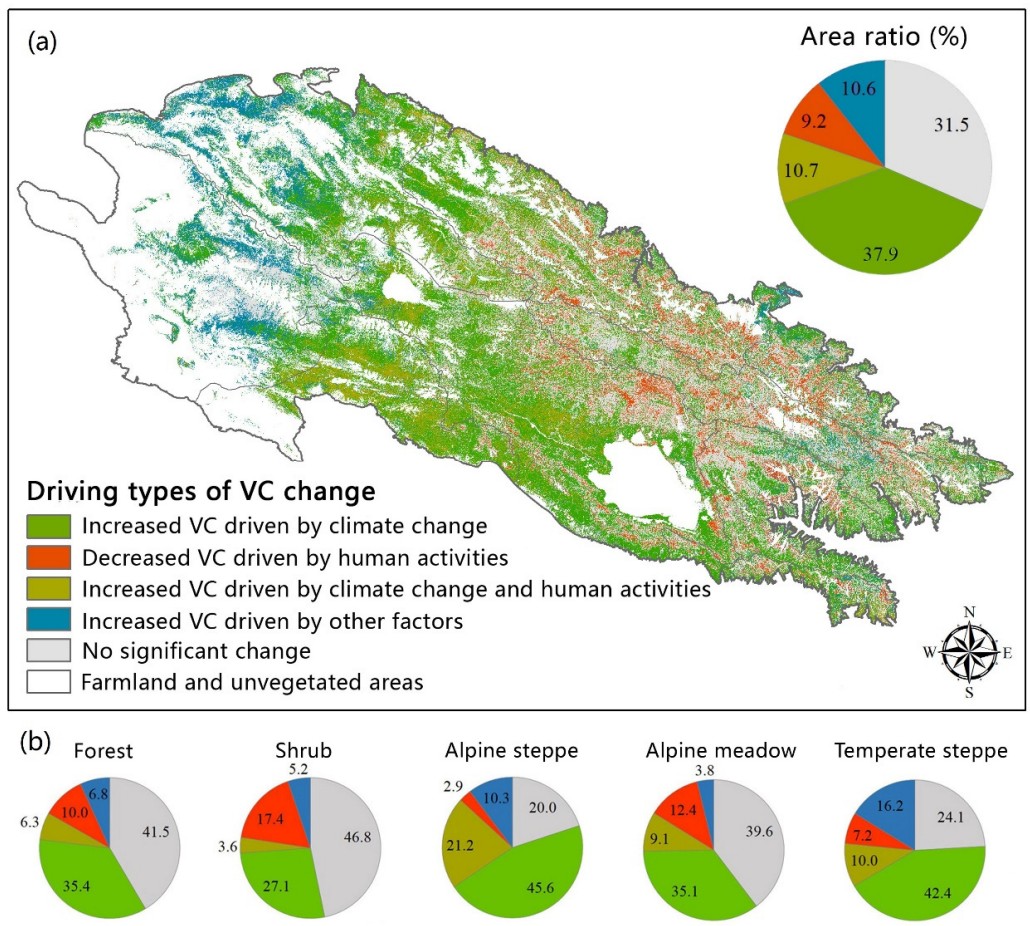

**Figure 6.** Climatically and anthropogenically driven pattern of vegetation coverage (VC) change (**a**) and percentages of VC change types under different vegetation types (**b**) in the QLM.

Based on the vegetation type (Figure 1a) and climatically and anthropogenically driven pattern of VC change (Figure 5a) in the QLM, we obtained the area proportion of VC change types for different vegetation types (Figure 6). Forests and shrubs are relatively stable ecosystems, with 41.5% and 46.8% of the regions having no significant change in VC, respectively. Affected by climate change, 35.4% of forests had significant improvement in VC, while another 6.3% of forests had significantly increased VC under the combined effects of climatic and anthropogenic factors. In addition, about 10% of forests had significantly reduced VC due to human activities. Shrubs were most obviously affected by human activities, with 17.4% of the area showing significant degradation due to human activities. Meanwhile, shrubs were least affected by climate change, with 30.7% of the areas exhibiting significant increases in VC due to the comprehensive impacts of climate change and human activities. The impact of climate change on alpine steppes was most significant, with 66.8% of the VC increase driven by climate change. Human activities had extremely low interference, with only 2.9% of the areas degraded by human activities. Affected by human activities, 12.4% of alpine meadows significantly degraded. In addition, caused by the comprehensive impacts of climatic and anthropogenic factors, approximately 48% of alpine meadows significantly increased in VC. The influence of climate change on temperate steppes was second only to that on alpine steppes, with 52.4% of the regions significantly increased in VC due to climate change and human activities. However, 7.2% of the areas also showed a significant decrease in VC caused by human activities.

## 4. Discussion

### 4.1. Impacts of Natural and Human Factors on Vegetation Coverage Change

The overall VC in the QLM improved remarkably from 1990 to 2018, which was largely attributed to climate change and ecological restoration. Our research showed that climate change played a leading role in the VC increase in the QLM, and the influence of precipitation was obviously greater than that of temperature. One possible reason is that the climate in the QLM is relatively dry, and vegetation growth is more sensitive to precipitation change [29]. In addition, active artificial restoration, such as afforestation and mine restoration, also played a significant role in improving VC in local areas [38,42]. The climate warming and humidifying has significantly increased VC in the QLM, but extensive vegetation degradation still exists. The results of the correlation analysis and residual analysis showed that human activities were the main factors causing vegetation degradation in the QLM (Table 2, Figure 6). Of all human activities, mining development and hydropower construction were the leading factors in the rapid deterioration of vegetation conditions [55,56]. Meanwhile, we found a very strong positive correlation between vegetation degradation and grazing intensity, which indicates that grazing activities may be one of the most important factors [57]. In addition, VC change was significantly negatively correlated with elevation and slope. That is, vegetation degradation tends to occur in areas with relatively low elevations and gentle slopes, which may be related to the strong human activities in these areas [58].

For vegetation types, forests, shrubs, and alpine meadows distributed in areas with superior natural conditions and intense human activities experienced a certain amount of degradation (Figures 2 and 6). A relevant study also showed that the alpine meadows, shrubs, and forests in the QLM were most affected by human interference due to the increase of tourism, overgrazing, and other disturbances [59]. In some areas, forests had a single-stand structure and underwent degradation due to drought, forest fires, and infection with pests and diseases [60]. In addition, research has found that overgrazing is the main factor causing a decrease in VC in shrubs, sparse forests, and young forests, especially in shrubs with VC ranging from 30% to 40% [61]. Therefore, it is still necessary to reduce grazing intensity in the future, especially for alpine meadows and shrubs with significant degradation. Meanwhile, it is particularly important to conduct ecological engineering in areas with good natural conditions and relatively dense populations to promote local vegetation restoration.

### 4.2. Limitations and Prospects

Consistent with previous studies [36,37], the areas of significant increase and decrease in VC were mainly distributed in the western and eastern QLM, respectively. However, there have been significant spatial differences in the VC change among diverse studies. On the one hand, this may be due to the different research periods and the significant discrepancies in VC change between different years. On the other hand, differences in data sources and resolution can also lead to differences in analysis results. The 30 m resolution VC data in this study come from reflectance data of red and near-infrared channels from Landsat5, Landsat8, and Sentinel 2 [62]. Although various datasets have been widely used in vegetation change analysis, there are large deviations between different sensors. Therefore, comparative analysis of multi-source data and multiple time scales may make the results more robust and accurate. Affected by human activities, land use types in the QLM have changed significantly from 1990 to 2018 [63], and vegetation types will also change accordingly. In this research, vegetation type data were extracted from the 1:1,000,000 Chinese Vegetation Atlas, which reflects the distribution of 11 vegetation type groups in China around the year 2000. Therefore, these data can only reflect the status of vegetation types in general, and it is difficult to accurately show the current distribution of various vegetation types. In addition, this study only analyzed the impacts of the two essential climatic factors, precipitation and temperature, on vegetation change. The direct and indirect effects of various climatic variables on vegetation dynamics are complicated.

Thus, further studies should be carried out by combining various climatic factors, and the interaction relationships of climatic factors should be identified.

**5. Conclusions**

This study analyzed the temporal and spatial characteristics of VC change in the QLM from 1990 to 2018 and recognized its main natural and anthropogenic influencing factors. Furthermore, we identified a climate-driven pattern of VC change and obtained the VC change zones driven by climate change and human activities. In addition, we also analyzed VC change intensity and types under different vegetation types. The main conclusions are as follows:

(1) VC in the QLM showed an overall fluctuating increase trend from 1990 to 2018. Areas with significant increases were mainly distributed in the central and western QLM, and regions with significant decreases were mostly located in the central and eastern QLM. The decrease in VC is more pronounced in areas with high annual precipitation, high VC, and high livestock density. For vegetation types, the VC of alpine steppes and temperate steppes increased significantly, while forests, shrubs, and alpine meadows deteriorated.

(2) The increase of VC in the QLM was primarily driven by climate change, and the effect of precipitation increasing was more obvious than that of temperature increasing. Among them, the precipitation-driven type was mostly distributed in the central high-altitude region and the eastern margin, the co-driven zone of temperature and precipitation was largely located in the central QLM, and the temperature-driven zone was relatively scattered. For vegetation types, alpine grasslands were the most affected by climate change, followed by temperate grasslands, forests, and alpine meadows.

(3) Vegetation degradation in the QLM was primarily attributed to anthropogenic factors. Areas where human activities had an obvious negative effect were largely distributed in the eastern part of the QLM, especially the Huangshui River Basin and the areas around Qinghai Lake. The regions with enhanced human disturbances were mainly distributed in the north–central QLM, mainly involving the Heihe River Basin, Datong River Basin and Shiyang River Basin. In terms of vegetation type, shrubs were most disturbed by human activities, followed by alpine meadows and forests.

**Author Contributions:** Conceptualization, Y.T.; data curation, X.W.; formal analysis, Y.T.; funding acquisition, M.S.; methodology, L.W.; project administration, M.S. and J.Z.; resources, C.W.; supervision, M.S. and J.Z.; validation, X.W.; visualization, L.W.; writing—original draft, Y.T.; writing—review and editing, C.W. and J.Z. All authors have read and agreed to the published version of the manuscript.

**Funding:** The research was funded by the Second Scientific Expedition to the Qinghai–Tibet Plateau (grant number 2019QZKK0405-05), the Guangdong Basic and Applied Basic Research Foundation (grant number 2022A1515110665), and the China Postdoctoral Science Foundation (grant number 2022M722522).

**Data Availability Statement:** All data used in this study are available from the author by request (tengym_simlab@163.com).

**Acknowledgments:** We are grateful to the editors and reviewers for their constructive work.

**Conflicts of Interest:** The authors declare no conflict of interest.

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
