# Peer review of "Characteristics of Vegetation Change and Its Climatic and Anthropogenic Driven Pattern in the Qilian Mountains"

_forests, doi:10.3390/f14101951_

Round 1
Reviewer 1 Report
The paper deals with an interesting theme highly topical in the context of the current World research trends. Authors present vegetation change and its climatic and anthropogenic driven pattern in the specific region Qilian Mountains (China). I appreciate especially the detailed analyses of data of NDVI change, as well as the interpretation of the obtained results. In order to meet the objective of the paper, the authors chose an adequate methodical apparatus based on the use of relevant geospatial and remote sensing data from 1990 and 2018 and modern geoinformatics equipment. Authors based on remote sensing and GIS techniques, analyzed the spatial and temporal changes of vegetation coverage in study area The results indicate that vlimate change played a leading role in the increase of vegetation coverage and the effect of precipitation increase was more obvious than that of air temperature increase. Vegetation degradation in the study area was mostly caused by human factors, and the areas where human activities had an obvious negative effect on vegetation change were largely distributed in the eastern part od mountains.
The title of the paper is acceptable and adequate and no major changes are necessary. I find the abstract acceptable and well structured. The manuscript has a sufficient scientific value and the information provided represents widening of knowledge. The conclusions are based entirely on the results and the methods used are adequate. The relation between the scientific value and the extent is acceptable. The language and style of the text are at an acceptable level. The tables and illustrations used in the paper are adequate; however I consider the number of references incomplete. The topic dealt with in the paper is also covered by other authors in papers.
I have no other remarks of a rather significant nature concerning the paper. The results are valuable and the scientific paper brings new original data. The manuscript is acceptable after minor revision with minor amendments required; no re-review is necessary. I recommend the paper for the print. The paper was of very good quality and it was a pleasure to read and review it.
So no elements that should be corrected:
Conclusion: any limitation of your research? So please add it.
I recommend amending the references. This issue is also covered by the newer papers from other authors. I recommend adding some papers into the references.
Figure 1, 3, 4 and 5 – especially the text of legend is very small.
As you see, there is not too much to correct according to my opinion.
Good luck in your future scientific work.
Author Response
Detailed response to Reviewer #1
Comment 1:
The paper deals with an interesting theme highly topical in the context of the current World research trends. Authors present vegetation change and its climatic and anthropogenic driven pattern in the specific region Qilian Mountains (China). I appreciate especially the detailed analyses of data of NDVI change, as well as the interpretation of the obtained results. In order to meet the objective of the paper, the authors chose an adequate methodical apparatus based on the use of relevant geospatial and remote sensing data from 1990 and 2018 and modern geoinformatics equipment. Authors based on remote sensing and GIS techniques, analyzed the spatial and temporal changes of vegetation coverage in study area The results indicate that climate change played a leading role in the increase of vegetation coverage and the effect of precipitation increase was more obvious than that of air temperature increase. Vegetation degradation in the study area was mostly caused by human factors, and the areas where human activities had an obvious negative effect on vegetation change were largely distributed in the eastern part of Qilian Mountains.
The title of the paper is acceptable and adequate and no major changes are necessary. I find the abstract acceptable and well structured. The manuscript has a sufficient scientific value and the information provided represents widening of knowledge. The conclusions are based entirely on the results and the methods used are adequate. The relation between the scientific value and the extent is acceptable. The language and style of the text are at an acceptable level. The tables and illustrations used in the paper are adequate; however I consider the number of references incomplete. The topic dealt with in the paper is also covered by other authors in papers.
I have no other remarks of a rather significant nature concerning the paper. The results are valuable and the scientific paper brings new original data. The manuscript is acceptable after minor revision with minor amendments required; no re-review is necessary. I recommend the paper for the print. The paper was of very good quality and it was a pleasure to read and review it.
Response to Reviewer’s comment No. 1:
At first, thank you very much for your affirmation of this study. We have revised this manuscript according to your advices, and we believe that the quality of the manuscript will be improved a lot based on your suggestions.
Comment 2:
Conclusion: any limitation of your research? So please add it.
Response to Reviewer’s comment No. 2:
Thank you for your comment. We have add the limitations in the manuscript:
“Consistent with previous studies [36, 37], the areas of significant increase and decrease in VC were mainly distributed in the western and eastern QLM, respectively. However, there were significant spatial differences in VC change among diverse researches. On the one hand, it may be due to the different research periods, and the significant discrepancies in VC change between different years. On the other hand, differences in data sources and resolution can also lead to differences in analysis results. The 30 m resolution VC data in this study comes from reflectance data of red and near-infrared channels from Landsat5, Landsat8, and Sentinel 2 [62]. Although various data sets have been widely used in vegetation change analysis, there are large deviations between different sensors. Therefore, comparative analysis of multi-source data and multiple time scales may make the results more robust and accurate. Affected by human activities, land use types in the QLM have changed significantly from 1990 to 2018 [63], and vegetation types will also change accordingly. In this research, vegetation type data was extracted from the 1: 1 000 000 Chinese Vegetation Atlas, which reflects the distribution of 11 vegetation type groups in China around 2000. Therefore, this data can only reflect the status of vegetation types in general, and it is difficult to accurately show the current distribution of various vegetation types. In addition, this study only analyzed the impact of the two essential climatic factors, precipitation and temperature, on vegetation change. The direct and indirect effects of various climatic variables on vegetation dynamics are complicated, and further studies should be carried out by combining various climatic factors, and the interaction relationships of climatic factors should be identified.”
Comment 3:
I recommend amending the references. This issue is also covered by the newer papers from other authors. I recommend adding some papers into the references.
Response to Reviewer’s comment No. 3:
Thank you for your suggestion. We have added several new papers in the references:
“29.Zhang, L.; Yan, H.; Qiu, L.; Cao, S.; He, Y.; Pang, G. Spatial and temporal analyses of vegetation changes at multiple time scales in the Qilian Mountains. Remote Sens-Basel 2021, 13(24), 5046.
30.Zhang, H.; Li, M.; Song, J. Analysis of driving factors of vegetation NDVI change in Qilian Mountain National Park based on geographic detector. Chin. J. Ecol. 2021, 40(08): 2530-2540.
31.Li, J.; Gong, C. Effects of terrain factors on vegetation cover change in National Park of Qilian Mountain. Bull. Soil Water Conserv. 2021, 41(03): 228-237.
45.An, J.; Niu, Y.; Che, Z.; Hao, H. Grey correlation analysis of water conservation function of typical vegetation types in the alpine region of the Qilian Mountains. J. Central South University Forest. Tech. 2023, (08), 93-101.
46.Hu, Z.; Song, X.; Tan, L.; Liu, H.; Wen, W. Spatio-temporal variation characteristics and its driving factors of NDVI at county scale for an inland arid grassland during 2001—2020. Bull. Soil Water Conserv. 2022, 42 (5), 213-221.
55.Qian, D.; Yan, C.; Xiu, L.; Feng, K. The impact of mining changes on surrounding lands and ecosystem service value in the southern slope of Qilian Mountains. Ecol. Complex. 2018, 36, 138–148.
56.Li, Y.; Li, Z.; Zhang, X.; Yang, A.; Gui, j.; Xue, J. Spatial and temporal changes in vegetation cover and response to human activities in Qilian Mountain National Park. Acta Ecol. Sin. 2023, 43 (1), 219-233.
59.Duan, Q.; Luo, L.; Zhao, W.; Zhuang, Y.; Liu, F. Mapping and evaluating human pressure changes in the Qilian Mountains. Remote Sens-Basel 2021, 13 (12), 2400.
60.Quan, J. The dilemma and countermeasures of natural forest protection in Qilian Mountain Nature Reserve. Mod. Horticult. 2023, 46 (10), 174-176.
61.Yuan, H.; Wang, L.; Guo, S.; Wang, S.; Jin, M.; Wang Y. Investigation and analysis of forest area change in Qilian Mountain National Nature Reserve in Gansu Province. For. Sci. Technol. 2022, (6), 65-70.
63.Fu, J.; Cao, G.; Guo, W. Land use change and its driving force on the southern slope of Qilian Mountains from 1980 to 2018. Chin. J. Appl. Ecol. 2020, 31(08): 2699-2709.”
Comment 4:
Figure 1, 3, 4 and 5 – especially the text of legend is very small.
Response to Reviewer’s comment No. 4:
Thank you for your comment. The text in the picture cannot be enlarged because there is no extra space. Therefore, we enlarged the whole picture in the manuscript.
Reviewer 2 Report
The study has evaluated changes in vegetation cover in the Qilian Mountains (QLM) in response to several climatic and non-climatic factors. Several remote sensing datasets are used to investigate long-term changes in VC. The results of the present study are supported by data selection and techniques. However, the current form of the manuscript needs major improvement. The specific comments are given below.
Major Comments:
1) Abstract: It can be rewritten as there are some unclear statement that exists
2) Introduction: It is not properly contextualized. For example, driving factors of vegetation changes such as solar radiation, soil moisture and nutrient conditions are never mentioned in the intro.
3) Objectives: As mentioned in L92 on social influencing factors, I don’t find any results presented in this part
4) Methods: Eq. 7 needs a better description. How VCreal is calculated and so on …
5) VC data was in 30 m or 1 km resolution. It is not clear from the data section.
6) Fig 3: How the line plot for a and b is done. Write in methodology, how these data are extracted from the spatial map. Which point was considered? Whether average of all areas was shown in the line plot ?
7) Fig 3 c and d: At what threshold of correlation value is used to define significant and highly significant? In the legend, it should mention in the caption
8) Fig 4: Same comment as mentioned for Fig 3c and 3d
9) Add limitations of the study under discussion. For instance, how these results will vary if other trend methods will be used (like Man-Kendall, Sens slope so on …). As long-term data is used, how these data are robust due to changes in sensors
Minor comments
· L24-26: "VC change was significantly positively correlated ...".
· How VC was significantly negatively correlated with VC?...Something wring in this statement
· L26-28: Rewrite it... (Climate change....)
· L136: It says VC data is in 30 m resolution where as in L140, it becomes 1 km. Clarify which one is used
· L224: use Capital for we
· L226: VC change with VC. Rewrite
moderate edit is required
Author Response
Detailed response to Reviewer #2
Comment 1:
The study has evaluated changes in vegetation cover in the Qilian Mountains (QLM) in response to several climatic and non-climatic factors. Several remote sensing datasets are used to investigate long-term changes in VC. The results of the present study are supported by data selection and techniques. However, the current form of the manuscript needs major improvement.
Response to Reviewer’s comment No. 2:
Thank you for your careful review of this manuscript. We have made corresponding changes in the manuscript according to your comments. Thanks to your suggestions, we believe that the quality of the manuscript will be greatly improved. Thank you very much again.
Comment 2:
Abstract: It can be rewritten as there are some unclear statement that exists
Response to Reviewer’s comment No. 2:
Thank you for your comment. We have rewritten the abstract and here is the new version:
“The Qilian Mountains (QLM) is an essential ecological security barrier in northwest China. Identifying the driven pattern of vegetation change is crucial for ecological protection and restoration in the QLM. Based on the high-resolution vegetation coverage (VC) data in the QLM from 1990 to 2018, the linear trend analysis was employed to examine the spatio-temporal dynamics of VC in the QLM, while correlation analysis was utilized to establish relationships between VC change and environmental factors. Multiple correlation and residual analysis were adopted to recognize the climatic and anthropogenic driven pattern of VC change. The results showed that VC in the QLM presented a remarkable upward trend in volatility from 1990 to 2018. The significant increase areas accounted for 59.32%, mainly distributed in the central and western QLM, and the significant decrease areas accounted for 9.18%, mostly located in the middle and eastern QLM. VC change showed a significant positive correlation with precipitation change and annual average temperature, while it exhibited a significant negative correlation with annual average precipitation, current VC level, livestock density, and slope.Climate change played a leading role in the increase of VC, and the impact of precipitation was significantly higher than that of temperature. Affected by climate change, VC of alpine steppes and temperate steppes increased most. Under human interference, VC decreased significantly in 9.2% of the region, of which shrubs fell the most, followed by alpine meadows and forests. This study can provide certain guidance for local ecological protection and restoration.”
Comment 3:
Introduction: It is not properly contextualized. For example, driving factors of vegetation changes such as solar radiation, soil moisture and nutrient conditions are never mentioned in the intro.
Response to Reviewer’s comment No. 3:
Thank you for your comment. We have added relevant contents to the manuscript according to your comment:
“Besides precipitation and temperature, solar radiation and wind speed are also important driving factors affecting vegetation change [29]. Moreover, elevation and soil type were the main factors affecting VC in the Qilian Mountains(QLM) [17, 30]. For elevation gradient, VC in the QLM increased first and then decreased with the increase of altitude [30]. From the long-term change, VC showed an increasing trend at low altitudes (below 3200m) and gentle slopes (below 15°), and a decreasing trend at high altitudes (above 3700m) and steep slopes (above 25°) [31]. For soil type, the black felt soil has abundant humus, which can provide rich nutrients for vegetation [17].”
Comment 4:
Objectives: As mentioned in L92 on social influencing factors, I don’t find any results presented in this part
Response to Reviewer’s comment No. 4:
Thank you for your comment. In this study, social influencing factors are distance from rural settlements, distance from roads, livestock density. However, after your reminder, we think it is more appropriate to change it to anthropogenic influence factors. We also change it in the manuscript.
Comment 5:
Methods: Eq. 7 needs a better description. How VCreal is calculated and so on …
Response to Reviewer’s comment No. 5:
Thank you for your comment. We have revised it in the manuscript according to your comments. The additions are as follows:
VCpredicted = a × T + b × P + c (7)
ε = VCreal - VCpredicted (8)
Where VCpredicted and VCreal are predicted VC value based on regression models and observed VC value based on remote sensing images, respectively; VCreal is the VC data with a resolution of 30m; a, b, and c are model parameters;
Comment 6:
VC data was in 30 m or 1 km resolution. It is not clear from the data section.
Response to Reviewer’s comment No. 6:
Thank you for your comment. In order to obtain a high-resolution spatial data of vegetation coverage change, we adopted a 30m-resolution vegetation coverage data. Further, we wanted to identify the continuous change trend of vegetation coverage in the Qilian Mountains from 1990 to 2018. However, there are only 7 phases of vegetation coverage data with 30m resolution (1990, 1995, 2000, 2005, 2010, 2015 and 2018). Therefore, annual vegetation coverage data with 1-km resolution were used in this study. For time trend, we calculated the vegetation coverage change in the whole Qilian Mountains, so the 1-km resolution data can also meet the requirements.
Comment 7:
Fig 3: How the line plot for a and b is done. Write in methodology, how these data are extracted from the spatial map. Which point was considered? Whether average of all areas was shown in the line plot?
Response to Reviewer’s comment No. 7:
Thank you for your comment. We have added relevant methodological descriptions to the manuscript:
“Further, based on the boundary data of the QLM, we extracted the average values of 1-km resolution annual VC , temperature, and precipitation year by year in ArcGIS, respectively.Then, we used the extracted data to create line plots, and obtained the change trends of VC, temperature, and precipitation in time series from 1990 to 2018.”
Comment 8:
Fig 3 c and d: At what threshold of correlation value is used to define significant and highly significant? In the legend, it should mention in the caption. Fig 4: Same comment as mentioned for Fig 3c and 3d.
Response to Reviewer’s comment No. 8:
Thank you for your comment. The significant and highly significant correlations are the situations under p < 0.05 and p < 0.01, respectively. We have supplemented it in the manuscript.
Comment 9:
Add limitations of the study under discussion. For instance, how these results will vary if other trend methods will be used (like Man-Kendall, Sens slope so on …). As long-term data is used, how these data are robust due to changes in sensors.
Response to Reviewer’s comment No. 9:
Thank you for your comment. We have add the limitations in the manuscript:
“Consistent with previous studies [36, 37], the areas of significant increase and decrease in VC were mainly distributed in the western and eastern QLM, respectively. However, there were significant spatial differences in VC change among diverse researches. On the one hand, it may be due to the different research periods, and the significant discrepancies in VC change between different years. On the other hand, differences in data sources and resolution can also lead to differences in analysis results. The 30 m resolution VC data in this study comes from reflectance data of red and near-infrared channels from Landsat5, Landsat8, and Sentinel 2 [62]. Although various data sets have been widely used in vegetation change analysis, there are large deviations between different sensors. Therefore, comparative analysis of multi-source data and multiple time scales may make the results more robust and accurate. Affected by human activities, land use types in the QLM have changed significantly from 1990 to 2018 [63], and vegetation types will also change accordingly. In this research, vegetation type data was extracted from the 1: 1 000 000 Chinese Vegetation Atlas, which reflects the distribution of 11 vegetation type groups in China around 2000. Therefore, this data can only reflect the status of vegetation types in general, and it is difficult to accurately show the current distribution of various vegetation types. In addition, this study only analyzed the impact of the two essential climatic factors, precipitation and temperature, on vegetation change. The direct and indirect effects of various climatic variables on vegetation dynamics are complicated, and further studies should be carried out by combining various climatic factors, and the interaction relationships of climatic factors should be identified.”
Comment 10:
L24-26: "VC change was significantly positively correlated ...". How VC was significantly negatively correlated with VC?...Something wring in this statement
Response to Reviewer’s comment No. 10:
Thank you for your comment. In this study, we want to know if vegetation coverage (VC) is more likely to change in areas with good vegetation conditions. Hence, we analyzed the correlation between VC change and current VC status. The results showed that VC change was significantly negatively correlated with current VC status. In other words, areas with significant vegetation degradation tend to be distributed in areas with high VC.
Comment 11:
L26-28: Rewrite it... (Climate change....)
Response to Reviewer’s comment No. 11:
Thank you for your comment. We have rewrite it in the manuscript.
“Climate change played a leading role in the increase of VC, and the impact of precipitation was significantly higher than that of temperature.”
Comment 12:
L224: use Capital for we
Response to Reviewer’s comment No. 12:
Thank you for your comment. We have revised it in the manuscript.
Round 2
Reviewer 2 Report
After reading the revised text, I find that the authors have addressed all the concerns and improved the manuscript.
moderate edit is required